# Tightly-Coupled Vehicle Positioning Method at Intersections Aided by UWB

**DOI:** 10.3390/s19132867

**Published:** 2019-06-28

**Authors:** Huaikun Gao, Xu Li

**Affiliations:** School of Instrument Science and Engineering, Southeast University, Sipailou 2, Xuanwu District, Nanjing 210096, China

**Keywords:** UWB, urban intersections, ARIMA, FCL, tightly-coupled, vehicle positioning

## Abstract

Reliable and precise vehicle positioning is essential for most intelligent transportation applications as well as autonomous driving. Due to satellite signal blocking, it can be challenging to achieve continuous lane-level positioning in GPS-denied environments such as urban canyons and crossroads. In this paper, a positioning strategy utilizing ultra-wide band (UWB) and low-cost onboard sensors is proposed, aimed at tracking vehicles in typical urban scenarios (such as intersections). UWB tech offers the potential of achieving high ranging accuracy through its ability to resolve multipath and penetrate obstacles. However, not line of sight (NLOS) propagation still has a high occurrence in intricate urban intersections and may significantly deteriorate positioning accuracy. Hence, we present an autoregressive integrated moving average (ARIMA) model to first address the NLOS problem. Then, we propose a tightly-coupled multi sensor fusion algorithm, in which the fuzzy calibration logic (FCL) is designed and introduced to adaptively adjust the dependence on each received UWB measurement to effectively mitigate NLOS and multipath interferences. At last, the proposed strategy is evaluated through experiments. Ground test results validate that this low-cost approach has the potential to achieve accurate, reliable and continuous localization, regardless of the GPS working statue.

## 1. Introduction

With the rapid development of transportation worldwide, it has become essential to realize accurate and reliable vehicle self-localization for many guidance and safety-related applications, such as intelligent transportation systems (ITS), advanced driver assistance systems (ADAS) and autonomous driving [1,2].

Traditional vehicle positioning, which utilizes low-cost onboard sensors, for instance, microelectromechanical Inertial Navigation System (MEMS-INS), electronic compass and odometer, can provide continuity and availability in some cases. Whereas, their accuracy degrades quickly over time on account of accumulated sensor errors. One comprehensive method to overcome this defect is integration of INS with the Global Positioning System (GPS). The fusion of INS and GPS performs well in open areas because of their complementary characteristics [3,4,5]. However, in GPS-denied environments such as tunnels, urban canyon or street junctions, the limited availability of satellite signals leads to a deteriorated positioning accuracy [6], which may dissatisfy most location-based applications in ITS.

Thus, in order to achieve accurate and uninterrupted vehicle positioning in typical urban scenarios, various sensor fusion approaches have been explored in the literature, which augment INS with other additional complementary sensors. In [7], a magnetometer was installed to aid INS for vehicle navigation and control. Zhang et al. proposed an integration of IMU with a digital compass and GPS in [8]. However, these sensors usually require to be specially installed and calibrated, which limits their applications. Sensor fusion between INS, GPS and vision [9,10] works quite well in certain circumstances, but they have a series of problems that are associated with real-time performance and light conditions. Similarly, Lidar-aided solutions [11,12] are vulnerable under adverse weather conditions. A digital 3D map can be used to improve vehicle positioning by sorting available satellites and mitigating the impact of multipaths [13]. Meanwhile, multipath mitigation techniques to classify invisible satellites by using an omnidirectional infrared (IR) camera comes to the fore in [14]. Despite the advantages of multipath mitigation, the lateral accuracy of those methods is limited due to satellite signals blocking the vehicle’s lateral direction.

Besides the previously mentioned supplementary method, wireless positioning technology such as radio frequency identification (RFID), dedicated short range communication (DSRC) and ultra-wide bandwidth (UWB) are recognized as potential assistant approach for positioning in a GPS-challenging environment. In [15], fusion strategy for vehicle positioning utilizing RFID and in-vehicle sensors have been demonstrated. Nima et al. proposed an instantaneous lane-level positioning using DSRC in [16]. However, in practical traffic scenarios such as urban intersections, radio signal propagation is easily obstructed by high buildings, trees or nearby vehicles; this method may suffer from multi-user interference, multipath effect and not line of sight (NLOS) propagation. Besides, vehicle positioning using UWB has also been investigated in [17,18,19]. Due to UWB’s excellent capability to penetrate obstacles and resolve multipath, UWB tech was introduced with great potential of achieving high-ranging accuracy.

Despite UWB having capacity of penetration as well as immunity to multipath, NLOS propagation has a high occurrence in harsh environments and it may significantly deteriorate positioning accuracy [20]. Vehicle positioning method-introduced UWB may suffer from NLOS problems. Several research studies [21,22,23] have been conducted to focus on NLOS identification and error mitigation in UWB’s raw measurements for dense indoor environments. However, the indoor environment is relatively stable and indoor targets are usually motionless or of slow-speed (compared with outdoor vehicles); methods developed for indoor environment may be not applicable for dynamic traffic scenarios.

To fuse information from multiple sensors, different filtering algorithms have extensively been studied and refined over time. Among them, the extended Kalman filter (EKF) [24] is the most widely implemented. Despite the EKF performing efficiently in many practical applications, its linearization process may suffer from divergence in a highly nonlinear system. Hence, the unscented Kalman filter (UKF) is introduced in [25]. Other nonlinear filtering algorithms have also been suggested; for instance, particle filter (PF) [26] and artificial neural networks [27]. However, these advanced filtering algorithms usually lead to a heavy computation load and are generally difficult for real-time implementation.

For the filtering methods discussed above, integration can be done in different frameworks. Generally, loosely-coupled integration solutions show less robustness compared to tightly-coupled methods proposed in [28,29,30], because their correction step cannot be carried out when the number of visible satellites is less than four.

Thus, aiming at tracking vehicles in typical urban scenarios as intersections, this paper proposes a tightly coupled integration strategy using UWB, GPS and MEMS-INS. In this strategy, the algorithms for both UWB raw measurements’ preprocessing and global fusion are developed to obtain a better performance. The main contributions of this paper are summarized as follows:(1)A UWB-based positioning method for land vehicles at typical urban intersections is proposed. Especially, in our proposal, an ARIMA model to address NLOS problem is presented and validated based on UWB experimental data. The gross errors in range measurements of NLOS nodes can be identified and corrected efficiently. To our knowledge, previous studies have rarely investigated the NLOS detection and mitigation for land vehicles in practical traffic scenarios.(2)A tightly-coupled adaptive fuzzy unscented Kalman filter algorithm (AF-UKF) is developed to realize global fusion. In implementation of the AF-UKF algorithm, a fuzzy calibration logic (FCL) is designed, which can adaptively adjust the dependence on each received UWB measurements to further mitigate the multipath and NLOS interferences.(3)Great precision improvement has been demonstrated through field tests while using low-cost GPS and MEMS-INS in this article.

This paper is organized as follows. Section 2 outlines the proposed positioning strategy using UWB. The NLOS mitigation algorithm is discussed in Section 3. Section 4 presents the tightly coupled fusion between GPS, INS and UWB. Experimental results are provided in Section 5. Section 6 is devoted to the concluding remarks.

## 2. Outline of the Proposed Positioning Strategy using UWB

Due to its ability to resolve multipath and penetrate obstacles, UWB technology offers the potential of achieving high ranging accuracy through time of arrive (TOA) measurements even in trash environments. The main advantages of utilizing UWB to realize vehicle positioning in road junctions are as follows:(a)Its extremely large bandwidth, usually between 3.6~10.1 GHz, makes it robust and more resistant to external interference.(b)Due to its impulse radio (IR) signal, UWB can penetrate obstacles.(c)It exhibits high ranging accuracy at decimeter level based on TOA measurements, and to some extent, at the centimeter level.(d)Apart from positioning, it can also be exploited for communication between vehicle and infrastructure, which is essential for most V2I applications in intersections.

In this paper, range measurements of UWB are obtained by the TOA method. As shown in Figure 1, in practice, the range measurements are not equal to the true distances, because of a number of effects, including multipath propagation, interference, not line of sight propagation and measurement noise.

Consider a positioning system as in Figure 2 where there are *N* fixed UWB anchors $p_{i}={\left[x_{i},y_{i},z_{i}\right]}^{T}$ (each anchor’s location is known when they are deployed), $\tilde{p}=[\tilde{x},\tilde{y},\tilde{z}]$ is the estimate of mobile node on vehicle, ${\hat{d}}_{i}$ is the measured range between the ith anchor and mobile node modeled as: (1)$${\hat{d}}_{i}=r_{i}\;+\;n_{i}+m_{i}+b_{i}=ct_{i}\;(i=1,2,\ldots ,N)$$
where $t_{i}$ is the TOA of the signal at the ith anchor, c is the speed of light, $r_{i}$ is the real distance between the mobile node and the ith anchor, $n_{i}$ is the measurement noise, which is usually modeled as additive white Gaussian noise ($n_{i}\sim N(0,\sigma_{i}^{2})$).

$m_{i}$ is the range error caused by multipath propagation. In enclosed spaces, $m_{i}$ can be a major component of localization bias. However, intersections are open areas compared to indoor environments. Owing to UWB’s ability to resolve multipath, $m_{i}$ is not taken into account in this paper.

$b_{i}$ is a positive range error introduced by the obstruction of direct path (as shown in Figure 3).
(2)$$b_{i}=\left\{ \begin{array}{ll} 0, & \mathrm{if}\;i\mathrm{th}\;\mathrm{anchor}\;\mathrm{is}\;\mathrm{LOS}\\ \delta_{i}, & \mathrm{if}\;i\mathrm{th}\;\mathrm{anchor}\;\mathrm{is}\;\mathrm{NLOS} \end{array}\right.$$

## 3. NLOS Mitigation Algorithm of UWB Anchors

As mentioned above, bias item $\delta_{i}$ is apparently larger in an NLOS situation compared to LOS propagation and leads to accuracy deterioration. Hence, an autoregressive integrated moving average model ARIMA (*p*,*d*,*q*) is constructed and used to tackle gross errors in TOA range measurements of NLOS anchors.

ARMA is the most common and effective method of modeling stationary time series. However, in the actual application, the vast majority of the time series is definitely nonstationary. In order to model nonstationary time series, Box and Jenkins brought forward the autoregressive integrated moving average (ARIMA) model. Generally speaking, ARIMA (*p*,*d*,*q*) consists of autoregressive model AR (*p*), moving average model MA (*q*) and autoregressive moving average model ARMA (*p*,*q*) models. In the context of this approach, the nonstationary series are required to perform d time differences until they transform into a stationary series before applying ARMA (*p*,*q*).

There are several methods that account for a series being stationary or not. The most popular is the augmented Dickey-Fuller test (ADF), which is applied to test the stationarity of UWB measurements in this paper.

For a non-stationary series of UWB range measurement d(k)(k=1,2,…,n), it needs to be differentiated before utilizing ARMA (*p*,*q*).

Identification of models usually relies on the analysis of the autocorrelation function (ACF) and the partial autocorrelation function (PACF). While the autocorrelation function measures the correlation between values in a time series separated by N, which represents the number of lags between these data, the partial autocorrelation function provides an indication in determining the number of lags in the AR model. Then, with the help of ACF and PACF, both p and q can be roughly acquired. The residual ACF and PACF are tools for model diagnostic checking.

After the above steps, an appropriate model ARIMA (*p*,*d*,*q*) is constructed and used to tackle gross errors. Suppose that the obtained ARIMA (*p*,*d*,*q*) model can be expressed as:(3)$$\begin{array}{ll} \hat{d}(k)= & a_{1}d(k-1)+a_{2}d(k-2)+\ldots +a_{p}d(k-p)+\\ & \varepsilon (k)-b_{1}\varepsilon (k-1)-b_{2}\varepsilon (k-2)-\ldots -b_{q}\varepsilon (k-q) \end{array}$$
where $a_{1},a_{2},\ldots ,a_{p}$ and $b_{1},b_{2},\ldots ,b_{q}$ are parameters of autoregressive component and the moving average component, respectively. ε(k),ε(k−1),…,ε(k−q) are Gaussian white noise series with mean zero and variance $\delta^{2}$. $\hat{d}(k)$ is the predication of d(k).

If $|\hat{d}(k)-d(k)|>\theta$, d(k) is identified as abnormal data of NLOS anchors and replaced by the median of $\hat{d}(k-p),\hat{d}(k-p+1),\ldots ,\hat{d}(k)$. The range precision of the consumer grade UWB modules used in this paper can achieve meter to sub-meter. Taking the application of vehicle positioning into consideration, the threshold θ is set to be 0.5 m in this paper.

The simplest way of NLOS mitigation is achieved by identifying and discarding the NLOS nodes and estimating the rover location by using LOS measurements. However, there is always the possibility of false identification, missed detection, and insufficiency of LOS nodes (available LOS nodes less than three), which degrade localization accuracy. Thus, a residual weighting algorithm is developed in this paper. 

Range measurement combinations Θ is given by:
(4)$$\Theta =\Theta_{los}\cup \Theta_{nlos}=\left\{S_{k}|k=3,4,\ldots ,N\right\}$$
where $S_{k}$ denotes the set of anchors for the kth combination. If N<3, Θ=∅.

For each set of combinations $S_{k}$, multilateration equations are as follows:(5)$$\left\{ \begin{array}{l} \sqrt{{\left(x-x_{1}\right)}^{2}+{\left(y-y_{1}\right)}^{2}+{\left(z-z_{1}\right)}^{2}}=b_{1}\\ \;\cdots \\ \sqrt{{\left(x-x_{i}\right)}^{2}+{\left(y-y_{i}\right)}^{2}+{\left(z-z_{i}\right)}^{2}}=b_{i}\\ \;\cdots \\ \sqrt{{\left(x-x_{k}\right)}^{2}+{\left(y-y_{k}\right)}^{2}+{\left(z-z_{k}\right)}^{2}}=b_{k} \end{array}\right.$$
where $\left(x_{i},y_{i},z_{i}\right)$ are the coordinates of UWB anchor nodes. $b_{1},\ldots ,b_{k}$ are the processed range measurements from the anchors in set $S_{k}$.

Compute an intermediate least square (LS) location estimate as follows:(6)$$\left\{ \begin{array}{l} R_{e}(p;S_{k})=A\cdot p_{k}-b_{S_{k}}\\ {\hat{p}}_{k}=\arg\;\underset{p}{\min}\left\{R_{e}(p;S_{k})\right\}\\ R_{e}(p;S_{k})=A\cdot {\hat{p}}_{k}-b_{S_{k}} \end{array}\right.$$
where $b_{S_{k}}=[b_{1},\ldots ,b_{k}]$ is the range measurement vector. $(A\cdot p_{k}-b_{S_{k}})$ is the matrix form of linearized LS multilateration equations. $R_{e}(p;S_{k})$ is the residual error when only the anchors in set $S_{k}$ are used for calculation. ${\hat{p}}_{k}$ is the LS estimation of the vehicle. The residual error can be normalized as:(7)$${\tilde{R}}_{e}({\hat{p}}_{k};S_{k})=R_{e}({\hat{p}}_{k};S_{k})/\left|S_{k}\right|$$
here, $\left|S_{k}\right|$ is the size of $S_{k}$.

The final location is obtained by weighting the intermediate location estimates with their corresponding normalized residual errors:(8)$$\tilde{p}=[\tilde{x},\tilde{y},\tilde{z}]=\frac{\sum_{k=1}^{N_{k}}{\hat{p}}_{k}{\left[{\tilde{R}}_{e}({\hat{p}}_{k};S_{k})\right]}^{-1}}{{\sum_{k=1}^{N_{k}}\left[{\tilde{R}}_{e}({\hat{p}}_{k};S_{k})\right]}^{-1}}$$

## 4. Tightly Coupled Localization Strategy Aiding by UWB

Since the preliminary positioning algorithm merely utilizing UWB has some deficiencies as discussed above, GPS, INS and other onboard sensors are introduced to enhance the positioning performance at the intersection.

In this paper, the tightly coupled strategy can feed raw GPS measurements (L1 and L2 pseudo-range) and UWB range to MEMS-INS through an UKF filter even when the number of visible satellites is three or fewer, thereby improving the performance of the navigation system in degraded GPS environments.

### 4.1. System Model

A nonlinear model involving the position, velocity, and attitude states based on a reduced inertial sensor system (RISS) is adopted in this paper [31]. Specifically, the RISS integrates the measurements from the vertically-aligned gyroscope and two horizontal accelerometers with velocity provided by wheel speed sensors. 

Then, the state model for tightly coupled integration is as follows:(9)$$X=[\varphi \;\lambda \;h\;v^{e}\;v^{n}\;v^{u}\;p\;r\;A\;sf^{od}\;b^{\omega }b^{GPS}\;d^{GPS}\;b^{UWB}{]}^{\prime }$$
where $[\varphi \;\lambda \;h\;v^{e}\;v^{n}\;v^{u}\;p\;r\;A{]}^{\prime }$ are latitude, longitude, height, the velocity component along east direction, the velocity component along north direction, the velocity component along the up direction, pitch angle, roll angle, azimuth angle, respectively;$sf^{od}$ is the scale factor error of the wheel speed; $b^{\omega }$ is the stochastic gyroscope drift, $sf^{od}$ and $b^{\omega }$ are generally modeled by Gauss-Markov model; $b^{GPS}$ and $d^{GPS}$ are GPS receiver clock offset and drift error; $b^{UWB}$ is a constant bias term of UWB range error, it can be modeled as a random constant process.

The receiver’s clock bias and drift errors of GPS can be modeled as:(10)$$\left[\begin{array}{l} \delta b_{k}^{GPS}\\ \delta d_{k}^{GPS} \end{array}\right]=\left[\begin{array}{l} 1\;\Delta t\\ 0\;1 \end{array}\right]\left[\begin{array}{l} \delta b_{k-1}^{GPS}\\ \delta d_{k-1}^{GPS} \end{array}\right]+\left[\begin{array}{l} \sigma_{b}\Delta t\\ \sigma_{d}\Delta t \end{array}\right]w_{k-1}^{GPS}$$

The measurements provided by the wheel speed sensor, the two accelerometers and the gyroscope comprise the control input:
(11)$$U=\;[v^{od}\;a^{od}\;f^{x}\;f^{y}\;\omega^{z}\;{]}^{\prime }$$
where $v^{od}$ is the vehicle speed derived from the vehicle’s wheel speed; $a^{od}$ is the vehicle acceleration derived from the vehicle’s wheel speed; $f^{x}$ and $f^{y}$ devote the transversal accelerometer measurement and forward accelerometer measurement, respectively; $\omega^{z}$ is the angular rate obtained from the vertically-aligned gyroscope.

The nonlinear system transition model about vehicle states is described as Equation (12).
(12)$$X_{k}=f(X_{k-1},U_{k-1},W_{k-1},T_{k-1})=\left[\begin{array}{c} \varphi_{k-1}+\frac{v_{k}^{n}}{R_{M}+h_{k}}\Delta t\\ \lambda_{k-1}+\frac{v_{k}^{e}}{(R_{N}+h_{k})\cos\varphi_{k}}\Delta t\\ h_{k-1}+v_{k}^{u}\Delta t\\ v_{k}^{od}\sin A_{k}\cos p_{k}\\ v_{k}^{od}\cos A_{k}\cos p_{k}\\ v_{k}^{od}\sin p_{k}\\ \sin^{-1}(\frac{f_{k}^{y}-a_{k}^{od}}{g})\\ -\sin^{-1}(\frac{f_{k}^{x}+v_{k}^{od}\omega_{k}^{z}}{g\cos p_{k}})\\ A_{k-1}-\omega_{k}^{z}\Delta t+(\omega^{e}\sin\varphi_{k-1})\Delta t+\frac{v_{k-1}^{e}\tan\varphi_{k-1}}{R_{N}+h_{k-1}}\Delta t\\ (1-\gamma_{w}\Delta t)\delta sf_{k-1}^{od}+\sqrt{2\gamma_{w}\sigma_{w}^{2}\Delta t}\\ (1-\beta_{z}\Delta t)\delta b_{k-1}^{\omega }+\sqrt{2\beta_{z}\sigma_{z}^{2}\Delta t}\\ \delta b_{k-1}^{GPS}+\delta d_{k-1}^{GPS}\Delta t\\ \delta d_{k-1}^{GPS}\\ \delta b_{k-1}^{UWB} \end{array}\right]$$
where $R_{M}$ is the meridian radius of curvature of the Earth. $R_{N}$ is the normal radius of curvature of the Earth. Δt is the sampling time. g is the acceleration of gravity. $\gamma_{w}$ is the reciprocal of the autocorrelation time for the scale factor of the wheel speed. $\beta_{z}$ is the reciprocal of the autocorrelation time for the gyroscope’s stochastic drift. $\sigma_{w}$ is the variance of the noise associated with the wheel speed. $\sigma_{z}$ is the variance of the noise associated with the gyroscope’s stochastic drift.

### 4.2. Observation Model

As discussed above, the observation information comes from low-cost GPS and UWB. The measurement vector Z is:
(13)$$Z=\;[\;\rho^{GPS,1}\cdots \rho^{GPS,M},\;d^{UWB,1}\cdots d^{UWB,N}{]}^{\prime }$$

Here, $\rho^{GPS,1}\cdots \rho^{GPS,M}$ are pseudorange measurements of available GPS; $d^{UWB,1}\cdots d^{UWB,N}$ are UWB range acquired from both LOS and NLOS anchors.

When the NLOS mitigation algorithm given in Section 3 is introduced, the quality of UWB measurements is significantly improved by ARIMA. The measurement vector can rewritten as Equation (13).
(14)$$Z=\;[\;\rho^{GPS,1}\cdots \rho^{GPS,M},\;{\hat{d}}^{UWB,1}\cdots {\hat{d}}^{UWB,N}{]}^{\prime }$$

The extended Kalman filter (EKF) has been widely used for INS and GPS integration. For system state model and measurement model above, the system input and measurement noises are assumed to be Gaussian with zero mean and their covariance matrices Q, Γ and R, respectively. Further, the corresponding EKF process can be divided into the following two phases:

(1) Prediction:(15)$$\hat{X}(k,k-1)=f(X(k-1),U(k-1),0,0)$$
(16)$$\begin{array}{ll} P(k,k-1)= & A(k,k-1)P(k-1)A^{T}(k,k-1)+\\ & B(k,k-1)\Gamma (k-1)B^{T}(k,k-1)+Q(k-1) \end{array}$$

(2) Update:(17)$$K(k)=P(k,k-1)•H^{T}(k)•{\left[H(k)P(k,k-1)H^{T}(k)+R(k)\right]}^{-1}$$
(18)$$\hat{X}(k)=\hat{X}(k,k-1)+K(k)[Z(k)-h(\hat{X}(k,k-1))]$$
(19)P(k)=[I−K(k)•H(k)]•P(k,k−1)
where I is an identity matrix, A and B are the Jacobian matrices of the system function f( ) with respect to X and U, and H is the Jacobian matrix of the measurement function h( ) with respect to X.

### 4.3. UKF Algorithm

For the model described above, both system and measurement models are nonlinear. Despite the EKF performing efficiently in many practical applications, its linearization process may suffer from divergence in a highly nonlinear system. Hence, to improve integration performance, an unscented Kalman filter (UKF) is introduced, which can directly use a nonlinear system and measurement models without any linearization. Generally, it can achieve a good balance between computational complexity and accuracy. 

The recursive procedure of UKF are carried out as follows:

(1) Calculate weights coefficient and sigma points:(20)$$\left\{ \begin{array}{ll} \omega_{0}^{\left(m\right)}=\frac{\eta }{n+\eta } & i=0\\ \omega_{0}^{\left(c\right)}=\frac{\eta }{n+\eta }+(1-\alpha_{1}^{2}+\alpha_{2}^{}) & i=0\\ \omega_{i}^{\left(m\right)}=\omega_{i}^{\left(c\right)}=\frac{1}{2(n+\eta )} & i=1,2,\ldots ,2n \end{array}\right.$$
(21)$$\left\{ \begin{array}{ll} \xi_{i}(k-1)=\bar{X}(k-1) & i=0\\ \xi_{i}(k-1)=\bar{X}(k-1)+{\left(\sqrt{\left(n+\eta )P(k\right)}\right)}_{i} & i=1,2,\ldots ,n\\ \xi_{i}(k-1)=\bar{X}(k-1)-{\left(\sqrt{\left(n+\eta )P(k\right)}\right)}_{i} & i=n+1,n+2,\ldots ,2n \end{array}\right.$$
where η is a scaling parameter, $\eta =\alpha_{1}^{2}(n+\alpha_{3})-n$. The constant $\alpha_{1}$ determines the spread of the sigma points around a mean of state X, and is usually set to a small positive value. The constant $\alpha_{3}$ is usually set to 0. $\alpha_{2}$ is used to incorporate prior knowledge of the distribution of X and is optimally set to 2 for Gaussian distributions, n is the dimension of state vector.

(2) Estimate the priori state $\hat{X}(k,k-1)$ and priori error covariance P(k,k−1), and then obtain the predicted observation by the unscented transform, the inputs of which are the sigma points and weights:(22)$$\xi_{i}(k,k-1)=f(\xi_{i}(k-1))\;i=0,1,\ldots ,2n$$
(23)$$\hat{X}(k,k-1)=\sum_{i=0}^{2n}\omega_{i}^{\left(m\right)}\xi_{i}(k,k-1)$$
(24)$$P(k,k-1)=Q(k)+\sum_{i=0}^{2n}\omega_{i}^{\left(c\right)}[\xi_{i}(k,k-1)-\hat{X}(k,k-1)]•{\left[\xi_{i}(k,k-1)-\hat{X}(k,k-1)\right]}^{T}$$
(25)$$\zeta_{i}(k,k-1)=h(\xi_{i}(k,k-1))$$
(26)$$\hat{Z}(k,k-1)=\sum_{i=0}^{2n}\omega_{i}^{\left(m\right)}\zeta_{i}(k,k-1)$$

(3) Calculate the UKF gain:(27)$$P_{ZZ}=R(k)+\sum_{i=0}^{2n}\omega_{i}^{\left(c\right)}[\zeta_{i}(k,k-1)-\hat{Ζ}(k,k-1)]•{\left[\zeta_{i}(k,k-1)-\hat{Z}(k,k-1)\right]}^{T}$$
(28)$$P_{XZ}=\sum_{i=0}^{2n}\omega_{i}^{\left(c\right)}[\xi_{i}(k,k-1)-\hat{X}(k,k-1)]•{\left[\xi_{i}(k,k-1)-\hat{X}(k,k-1)\right]}^{T}$$
(29)$$K(k)=P_{XZ}•P_{ZZ}{}^{-1}$$

(4) Update the system state and error covariance:(30)$$\hat{X}(k)=\hat{X}(k,k-1)+K(k)[Z(k)-\hat{Z}(k,k-1))]$$
(31)$$P(k)=P(k,k-1)-K(k)P_{ZZ}K{\left(k\right)}^{T}$$

### 4.4. AF-UKF Algorithm

As mentioned in Section 2, when UWB is used in harsh environments such as urban intersections, NLOS situations inherently have a high occurrence. NLOS and multipath propagation of UWB signal may significantly degrade positioning precision.

Although the ARIMA model is employed to identify and correct the gross errors in range measurements of the NLOS anchors, there are unsolved issues, such as multipath interference and scale factors of UWB, which may limit the performance of the fusion algorithm.

Some features of RF signals may indicate the quality of UWB measurement. As a localized communication tech, UWB range precision starts to deteriorate when the mobile node is far from the reference. Comparing with the neighborhood nodes, those remote references are less reliable. Besides, the antenna orientation may affect signal propagation as well.

In order to achieve accurate and reliable positioning at intersections, we can take full advantage of the above-mentioned features to determine the degree of dependence on the measurement of each received UWB signal, However, how to utilize the features above to ascertain the dependence degree on the measurement of each received UWB signal is a nonlinear process with some uncertainty and fuzziness, rather than a strict or rigorous process. Thus, it is reasonable to adopt fuzzy logic to model this decision process based on the features of the received UWB signals.

Hence, in this article, the fuzzy calibration logic (FCL) [30] here is designed using two inputs: $d_{i}$ and $\alpha_{i}$, where $d_{i}$ denotes the distance between the mobile node and the ith reference node. $\alpha_{i}$ is the azimuth difference between the reference node and the vehicle.

The input $d_{i}$ is described by three membership functions: Remote (R), Medium (M) and Nearby (N). Besides, two membership functions, Large (L) and Small (S) are used to describe input $\alpha_{i}$.

Then, for the output, five membership functions are defined as A, B, C, D and E, which denote the grade of output precision.

The fuzzy membership functions for two inputs and one output are shown in Figure 4.

The fuzzy reasoning rules are mainly based on prior practice. Table 1 gives the complete fuzzy rule base.

The fuzzy reasoning rules are mainly based on prior practice. Table 1 gives the complete fuzzy rule base.

The model adjustment coefficient $w_{i}$ is set mainly according to the knowledge or experience about the observation noise and is then adjusted many times by trial-and-error to obtain better estimation performance.

For each UWB reference node, the model adjustment coefficient $w_{i}$ can be determined utilizing the FCL above. When all UWB signals are judged, the noise covariance matrix of UWB measurements can be obtained from Equation (31).
(32)$$R_{}^{UWB}=diag[{\left(w_{1}R_{0}\right)}^{2}\;{\left(w_{2}R_{0}\right)}^{2}\ldots {\left(w_{i}R_{0}\right)}^{2}\ldots {\left(w_{N}R_{0}\right)}^{2}](i=1,2,\ldots ,N)$$
where $R_{0}$ is the basic noise covariance obtained by the prior knowledge and statistic trials. In practice, several static tests can be carried out to get the basic noise covariance. N is the number of UWB reference nodes. Combining statistic results with priori knowledge, we get the preliminary covariance matrix of the measurement noise of GPS and UWB.

Then, the whole adaptive noise covariance matrix R can be obtained as follows:
(33)$$R=diag[R_{}^{GPS}\;R_{}^{UWB}]$$

Thus, the sequential prediction steps and update steps are carried out using an UKF.

For clarity, we summarize the proposed adaptive fuzzy UKF (AF-UKF) algorithm briefly as follows:

Step (1) Calculate weights coefficient and sigma points according to Equations (20) and (21);

Step (2) Estimate the priori state and its error covariance and then calculate the predict observation based on Equations (22)~(26);

Step (3) Obtain the whole adaptive noise covariance matrix by Equations (32) and (33);

Step (4) Calculate the UKF gain according to Equations (27)~(29);

Step (5) Update the system state and error covariance using Equations (30) and (31).

## 5. Experimental Results

To verify the effectiveness of the proposed positioning solution, a series of experiments have been conducted at typical intersections in the Southeast university campus, as shown in Figure 5.

In the experiment, a consumer-grade GPS receiver (NovAtel C260-AT, NovAtel Inc, Calgary, AL, Canada) and low-cost MEMS-based inertial sensors (VG440CA-200 IMU, MEMSIC Inc, Andover, MA, USA) were equipped. In addition, wheel speed is acquired through an onboard CAN network.

Sensor accuracies (1σ) are 3 m and 0.05 m/s for the GPS position and GPS velocity, respectively; 0.1 m/s^2^ for the accelerometers, 0.2 °/s for the yaw rate sensor, and 0.05 m/s for wheel speed sensors. Moreover, an accurate NovAtel SPAN-CPT system was used as a reference for performance evaluation.

The selected test route included several driving scenes such as straight roads, curved roads, and intersections. Besides, a series of typical driving maneuvers, such as lane changes, accelerating, and decelerating, were conducted according to actual driving situations. During the experiment, all sensor data were collected, and then the positioning methods were evaluated using the logged data.

### 5.1. The Deployment of UWB Anchor at Typical Urban Intersections

In this paper, several low-cost UWB range modules (RK-101) based on 802.15.4a are used. The measurement errors(1σ) are 0.5 m under LOS conditions. The maximum working distance of RK-101 UWB module is about 200 m; however, the range accuracy deteriorates when the distance between two UWB nodes is over 100 m.

UWB anchors are deliberately deployed at the intersection as shown in Figure 6a: both sides of the street have an anchor node. In a roundabout intersection, it is quite easy to install a UWB module on road central infrastructures. The central node has less possibility of suffering from an NLOS situation. Thus, one node is placed at the center of the junction.

Note that the number of UWB anchors depends on the needs of the actual application. In our test, there are a total of nine anchors.

### 5.2. Performance of NLOS Mitigation Algorithm

Firstly, to evaluate the performance of the NLOS identification algorithm discussed in Section 3, straight line, curve line and comprehensive driving tests have been carried out, respectively. The driving test situations include acceleration, deceleration, and uniform motion under different vehicle speed conditions. For brevity, only one test is shown here as an example because similar conclusions can be reached from the other tests. The traditional residual weight least-squares (RWLS) method is investigated for multilateration. Figure 7, Figure 8 and Figure 9 show the trajectories of straight line, right turn and comprehensive driving test, respectively.

Owing to the deliberate deployment of UWB anchor nodes, the vehicle can be located integrally at the intersection, regardless of GPS. However, it can be seen from Figure 7 and Figure 8 that the blue trajectories (RWLS method without NLOS detection and mitigation) are deviated from the reference at the start and the end. The positioning errors are large when the vehicle drove into or drove away from the intersection. It indicates that NLOS propagation has a high occurrence when the vehicle is far from the center of the intersection.

The statistics of Euclidean distance errors (horizontal positioning errors) in Trajectory I and Trajectory II, which include maximum and root-mean-square (RMS), are summarized in Table 2. The low-cost GPS is the most widely used vehicle positioning sensor with accuracies (1σ) of about 3 m for position. From Table 2, we can find that the positioning accuracy of UWB without NLOS mitigation is approximately the same as that of low-cost GPS. However, the maximum positioning error is large and it may dissatisfy most location-based applications at intersections.

From Figure 7, Figure 8 and Table 2, it can be determined that positioning performance is obviously improved when an appropriate model ARIMA (during the experiments, the parameters of the ARIMA model are: *p* = 2, *d* = 1, *q* = 1) is constructed and used to tackle gross errors in measurements of NLOS anchors.

Then, a comprehensive test scenario is investigated in Test III. The trajectory and Euclidean distance errors are shown in Figure 9 and Figure 10, respectively. Table 3 illustrates the statistics of Euclidean distance errors (horizontal positioning errors) in Trajectory III, in which every possible driving operation is conducted, such as driving straight, right turn, left turn or “u” turn.

From the experimental results, we can find that the performances are poor when raw measurements of UWB are directly utilized. Owing to the ARIMA algorithm used in range measurement preprocessing, the maximum error is significantly decreased. For the RMS value of the Euclidean distance error, the ARIMA method achieves improvements in accuracy of about 35%.

In short, the results of Test I, II, III demonstrated that the ARIMA method proposed in this paper is suitable for NLOS error identification and correction for UWB in practical traffic scenarios.

### 5.3. Performance of Tightly Coupled Fusion Positioning

Note that the preliminary positioning algorithm merely utilizing UWB has some deficiencies as discussed above. In order to validate the performance of the tightly coupled AF-UKF method proposed in Section 4, further test is carried out. Figure 11 illustrates the positioning results of two methods.

Table 4 illustrates the statistics of Euclidean distance errors (horizontal positioning errors) in Trajectory IV.

In experiments IV and V, the tightly coupled strategy feed raw GPS measurements (L1 and L2 pseudo-range) and UWB range to MEMS-INS through an UKF filter even, when the number of GPS satellites are three or fewer, thereby verifying the performance improvement of the multi-sensor fusion method in degraded GPS environments.

As exhibited in Table 4, for the RMS value of the Euclidean distance error, an accuracy improvement up to 14% is acquired by the tightly-coupled AF-UKF fusion method when only two GPS satellites’ raw measurements are utilized. The result demonstrates that the proposed fusion method can remarkably improve positioning accuracy regardless of the GPS working status.

Test V is investigated to evaluate the effect of the fuzzy calibration logic (FCL) designed in Section 4.4. The test scenario contains every possible driving operation at the intersection, such as driving straight, right turn, left turn or “u” turn, as shown in Figure 12. The Euclidean distance errors of the three methods (RWLS, UKF and AF-UKF) investigated in the comprehensive test are displayed in Figure 13.

Table 5 illustrates the statistics of Euclidean distance errors (horizontal positioning errors) in Trajectory V. As can be seen from Figure 13 and Table 5, the AF-UKF method achieved better positioning performance than the other two methods.

The FCL may adaptively adjust the dependence on each received UWB measurement. Compared with the UKF method, AF-UKF provides a significant performance improvement, e.g., over 37%. This can be attributed to the fact that the proposed fuzzy calibration logic in AF-UKF can effectively alleviate the impact of NLOS and multipath propagations and improve the reliability of UWB measurements.

In a word, the comprehensive test V result demonstrated that the fusion strategy in Section 4.4 has the potential to achieve accurate, reliable, continuous and integrated line-level localization, regardless of the GPS’ working statue.

The test results above were acquired using the nine UWB reference nodes. Taking cost reduction into account, we also demonstrate the performance under different numbers and combinations of UWB anchors. UWB combination set Π is given by:(34)$$\Pi =\left\{G_{n}|n=3,4,\ldots ,9\right\}$$
where $G_{n}$ denotes the set of anchors achieving the best positioning performance among all of the $C_{N}^{n}$ combination of n anchors. Table 6 gives the minimum RMS errors from $G_{3}$ to $G_{9}$ in trials as test V.

This test indicates that the number of UWB anchors is essential to precise positioning at intersections. Positioning accuracy may degrade when the nodes are insufficient. However, owing to the AF-UKF fusion strategy used in this paper, accuracy deterioration is acceptable even if only three anchors are deployed.

## 6. Conclusions

UWB is recognized as a potential approach for positioning in a GPS-denied environment. In this paper, a positioning strategy using UWB, low-cost GPS and MEMS onboard sensors is proposed, aimed at tracking vehicles in typical urban scenarios (such as intersections).

NLOS situations inherently have a high occurrence in practical scenarios. Hence, we present an ARIMA model to address NLOS problems for vehicle positioning in urban intersections using TOA range measurements. A tightly coupled integration of GPS pseudorange measurements and UWB range is then developed to mitigate localization bias in NLOS situations. Furthermore, an FCL algorithm is designed and introduced to adaptively adjust the dependence on each received UWB measurement to effectively mitigate the NLOS and multipath interferences.

The proposed AF-UKF strategy was evaluated through typical experiments. Ground test results show that this low-cost method has the potential to achieve accurate, reliable, continuous and integrated localization, regardless of GPS’ working statue. However, it should be noted that the multipath propagation of UWB signal in harsh environments and scale factor of UWB range measurement are not taken into account in this paper. Our future work will be concerned with how to address those problems.

## Figures and Tables

**Figure 1 sensors-19-02867-f001:**
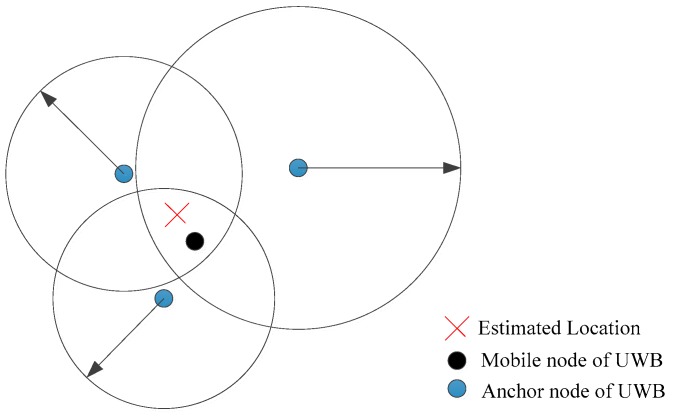
Illustration of UWB positioning based on ranging.

**Figure 2 sensors-19-02867-f002:**
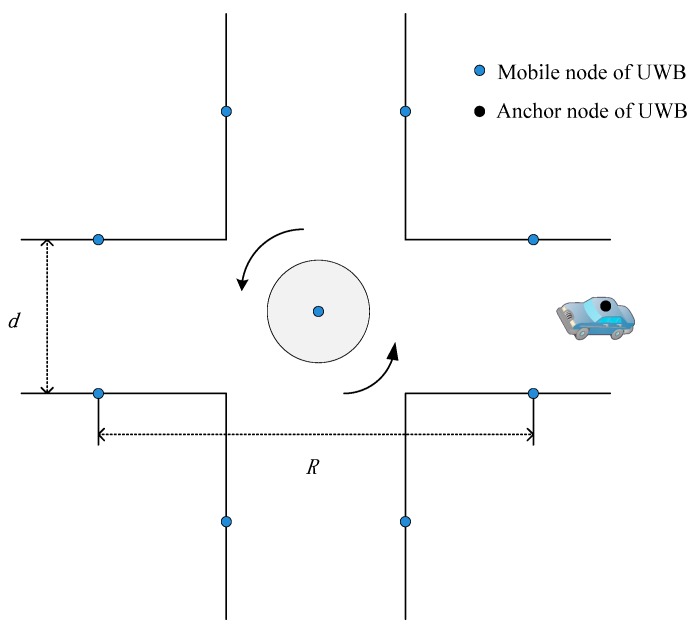
Illustration of UWB deployment at typical roundabout intersection.

**Figure 3 sensors-19-02867-f003:**
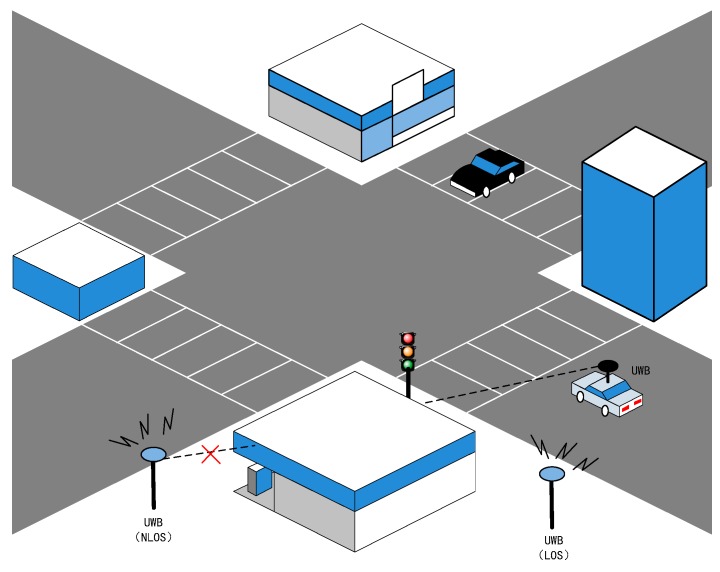
Typical NLOS situation in intersection scenarios.

**Figure 4 sensors-19-02867-f004:**
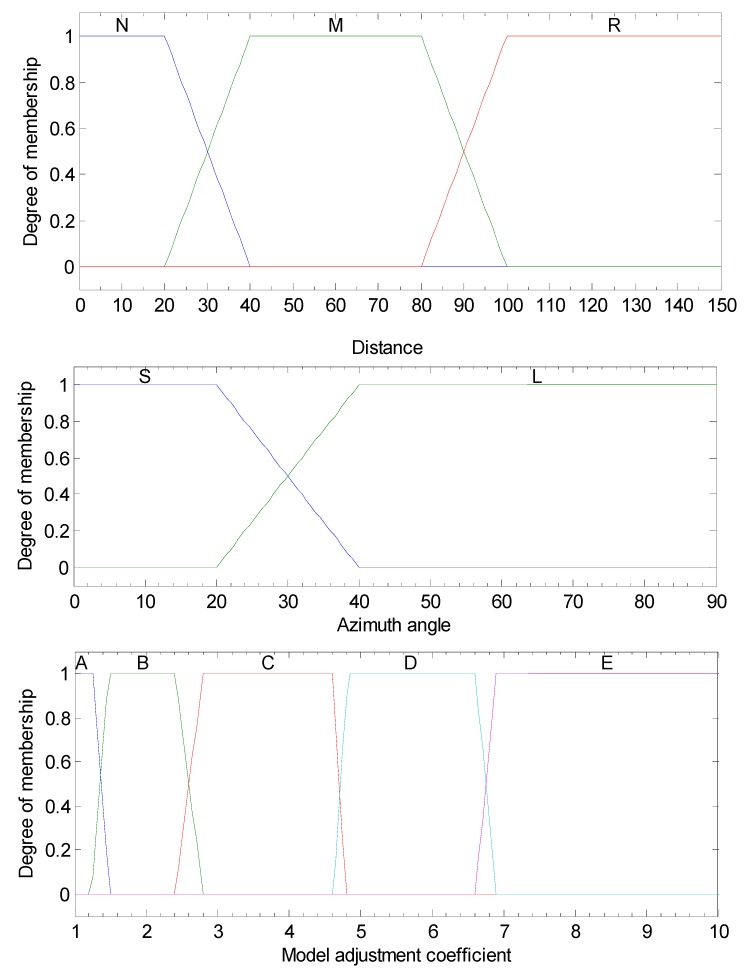
Membership functions for $d_{i}$ and $\alpha_{i}$.

**Figure 5 sensors-19-02867-f005:**
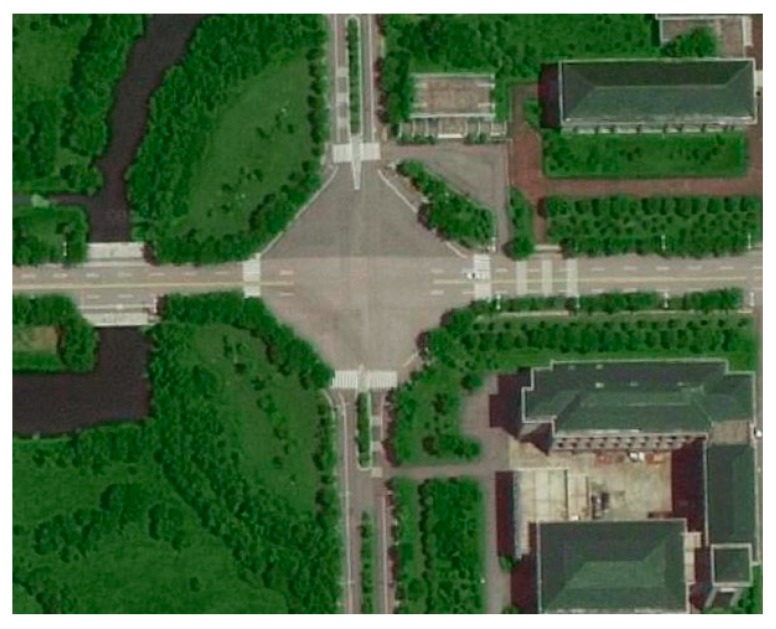
Bird’s-eye view of the intersection in ground test.

**Figure 6 sensors-19-02867-f006:**
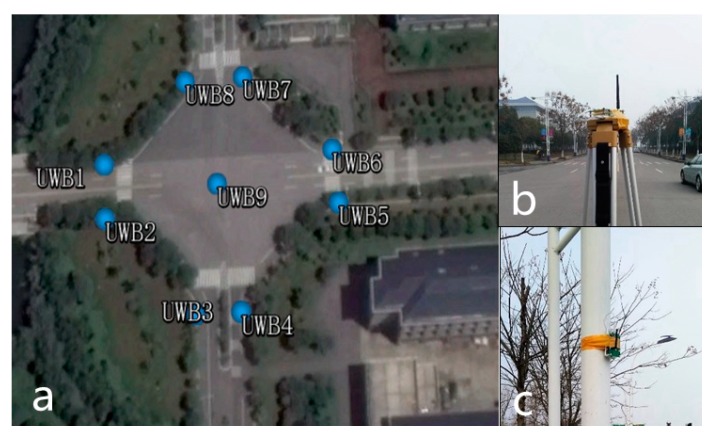
(**a**) Deployment of UWB anchors at the intersection: (**b)** centre anchor and (**c)** roadside anchor.

**Figure 7 sensors-19-02867-f007:**
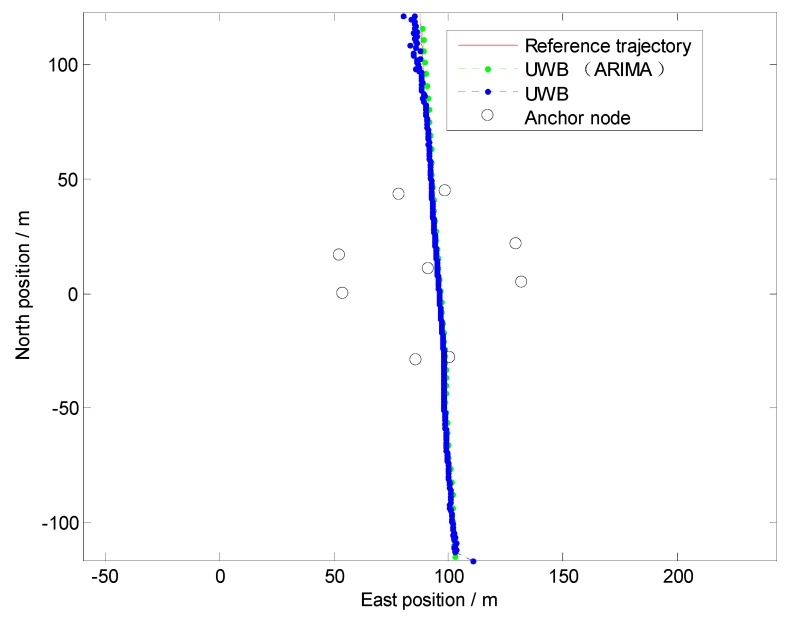
Positioning results in Trajectory I (Straight Line).

**Figure 8 sensors-19-02867-f008:**
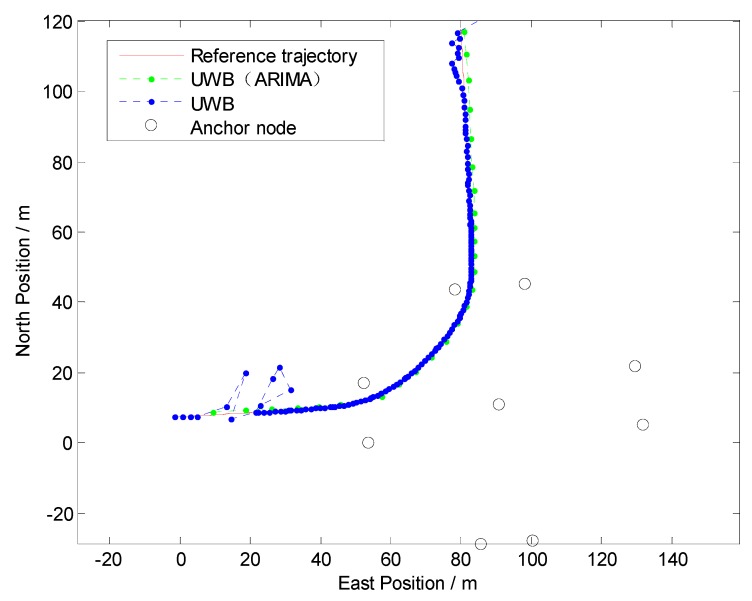
Positioning results in Trajectory II (Right Turn).

**Figure 9 sensors-19-02867-f009:**
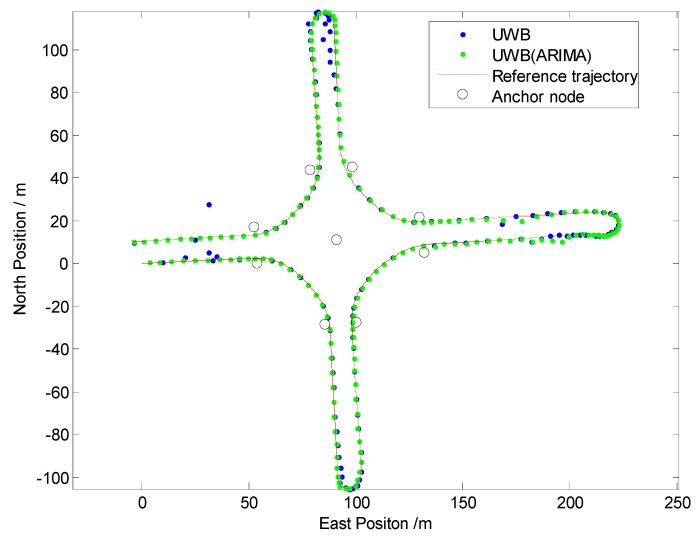
Positioning results in Trajectory III.

**Figure 10 sensors-19-02867-f010:**
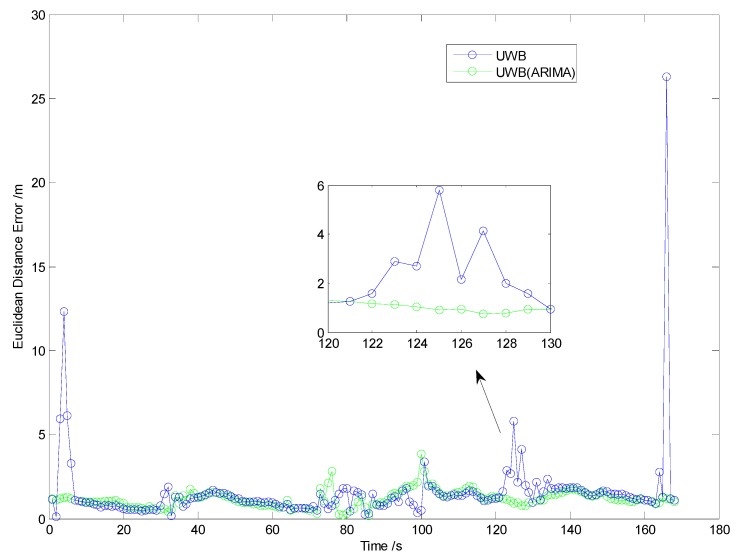
Euclidean distance error in Trajectory III.

**Figure 11 sensors-19-02867-f011:**
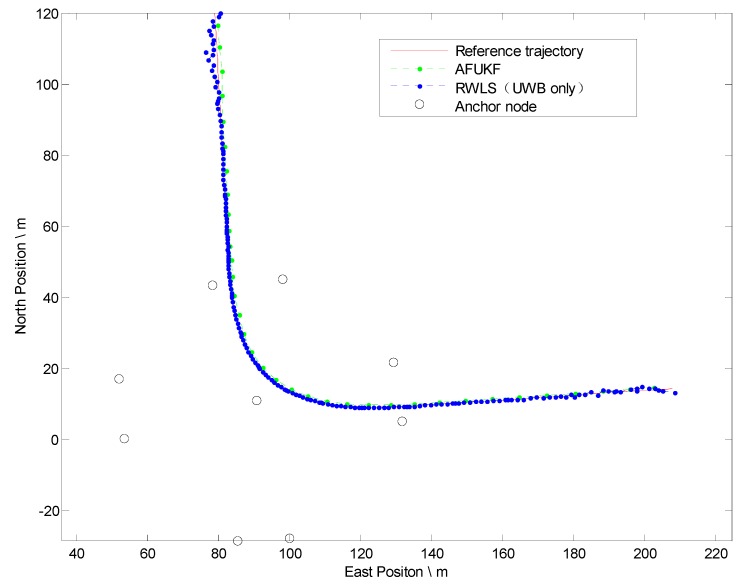
Positioning results in Trajectory IV (Left Turn).

**Figure 12 sensors-19-02867-f012:**
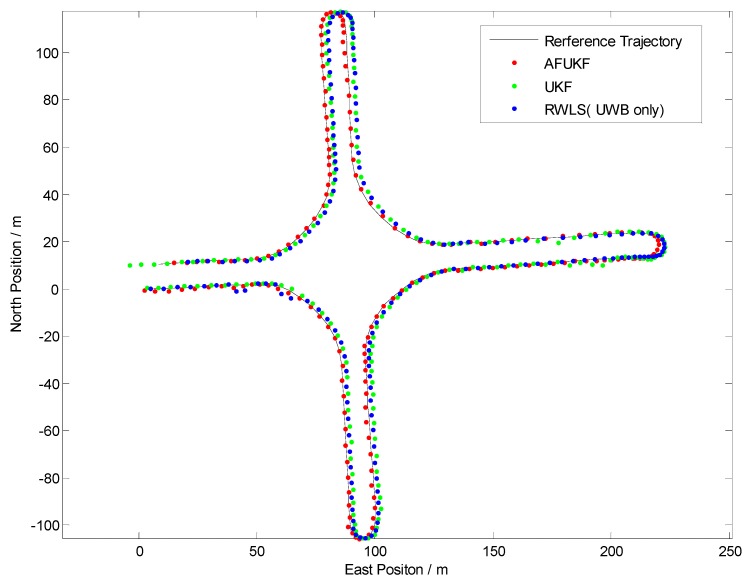
Positioning results in Trajectory V (Comprehensive Test).

**Figure 13 sensors-19-02867-f013:**
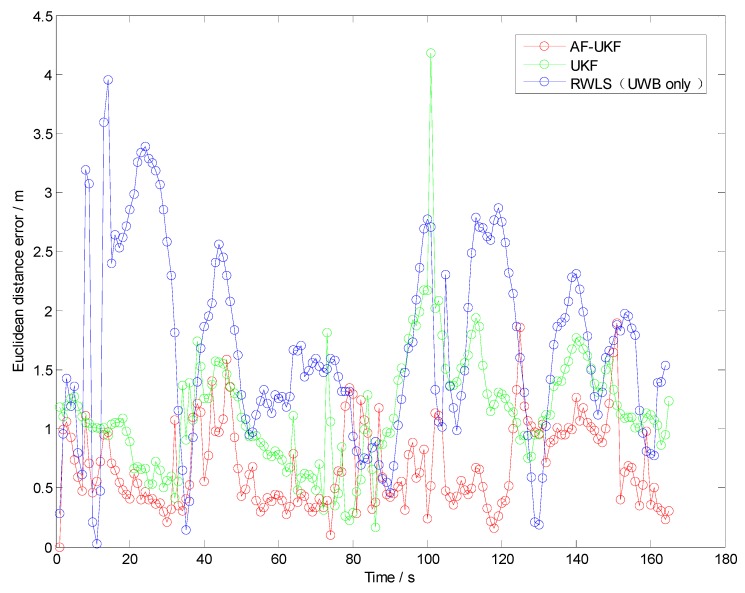
Euclidean distance error in Trajectory V.

**Table 1 sensors-19-02867-t001:** Fuzzy rules for proposed FCL.

Output Precision	$d_{i}$	$\alpha_{i}$
A	N	L
B	N	S
C	M	L
D	M	S
E	R	L
	R	S

**Table 2 sensors-19-02867-t002:** Statistics of Euclidean Distance Errors in two trajectories (m).

Trajectory	UWB		UWB(ARIMA)	
	Max	RMS	Max	RMS
I	15.62	2.85	6.72	1.74
II	18.26	2.92	7.62	1.92

**Table 3 sensors-19-02867-t003:** Statistics of Euclidean Distance Errors in Trajectory III (m).

Trajectory	UWB		UWB(ARIMA)	
	Max	RMS	Max	RMS
**III**	26.3	2.81	5.79	1.82

**Table 4 sensors-19-02867-t004:** Statistics of Euclidean Distance Errors in two trajectories (m).

Trajectory	RWLS (only UWB)		Tightly-Coupled AF-UKF	
	Max	RMS	Max	RMS
IV	5.75	1.78	4.86	1.52

**Table 5 sensors-19-02867-t005:** Statistics of Euclidean Distance Errors in trajectory V (m).

Trajectory	RWLS (UWB only)		UKF		AF-UKF	
	Max	RMS	Max	RMS	Max	RMS
V	3.95	1.86	4.08	1.51	1.91	0.95

**Table 6 sensors-19-02867-t006:** The minimum RMS errors from $G_{3}$ to $G_{9}$.

Combination Set	RMS	
	RWLS (UWB only)	AF-UKF
$G_{9}$	1.86	0.95
$G_{8}$	1.81	0.90
$G_{7}$	1.96	0.92
$G_{6}$	2.35	1.15
$G_{5}$	2.69	1.31
$G_{4}$	2.82	1.42
$G_{3}$	3.22	1.63

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
