# Peer review of "Tightly-Coupled Vehicle Positioning Method at Intersections Aided by UWB"

_sensors, 2019, doi:10.3390/s19132867_

Reviewer 1 Report

In order to may to track vehicle in typical scenarios as urban intersections, a positioning strategy utilizing ultra-wide band(UWB) and low-cost microelectron-mechanical (MEMS) onboard sensors is proposed.

I find that  the abstract and the conclusion include the important points of the paper but the both need to be reduced.

The figure and table captions are good presented and are accurate enough.

I miss the comparison between this method and the other published works.

Author Response

                                        Comments from Reviewer 1 and Responses

Comment: In order to may to track vehicle in typical scenarios as urban intersections, a positioning strategy utilizing ultra-wide band(UWB) and low-cost microelectron-mechanical (MEMS) onboard sensors is proposed. I find that the abstract and the conclusion include the important points of the paper but the both need to be reduced.

Response: Thanks a lot for your positive and encouraging comments. We have already reduced the abstract and the conclusion in the revised manuscript to make them concise and clear.

Comment: The figure and table captions are good presented and are accurate enough. I miss the comparison between this method and the other published works.

Response: Special thanks to you for your insightful comments. Compared with the other published works, we concentrated on how to address NLOS problem and achieve lane-level positioning at urban intersections using low-cost sensors (MEMS-INS and commercial grade GPS).

The major contribution of our work is the proposed ARIMA algorithm for NLOS detection and mitigation. The previous studies have rarely investigated the NLOS problem for UWB in practical traffic scenario before. In order to validate the performance of this method, we have investigated one of a conventional and typic range estimation-based NLOS detection and mitigation algorithms in reference [21] (namely “Survey of NLOS identification and error mitigation problems in UWB-based positioning algorithms for dense environments”). UWB range measurement processing result is shown in the followed figure (See in the notes file).

Compared with the conventional range estimation-based method, the ARIMA algorithm show the benefit of prompt NLOS detection. Besides, the mitigation performance is better than the conventional range estimation-based method. That is why we develop the ARIMA algorithm to address NLOS problem in dynamic traffic scenarios in this paper. Thanks again for your thorough consideration.

Reviewer 2 Report

In this work, the authors propose a vehicle localization scheme based on sensor fusion of  UWB/GPS/INS. To mitigate the effect of NLOS for UWB systems, the authors develop methods using ARIMA and fuzzy calibration logic. Most parts of the paper are well written. The authors need to address the following concerns in their revisions.

1.  Please explain what distinguishes this paper with the previous work [17]-[19].

2. The authors mentioned that the problem of NLOS identification and error mitigation has been studied for UWB in dense indoor environment, but not for outdoors. In general, indoor scenario is more challenging. Then why are the methods developed for indoors not applicable for outdoors?

3. Fix "trash environment" in page 3.

4. Please explain why the stationary assumption is essential for NLOS mitigation.

5. Write the explicit mathematical form for the residual error R_e(o;S_k) in (5).

6. What does "UWB bias error" mean in page 7.

7. Denote the first 9 terms in [8] one by one.

8. Is there any reference for the system transition model in (11)?

9. Explain in practice how to get the covariance matrix for the measurement noise of GPS and UWB in (16)-(18).

10. In Figure 10, the positioning error is very large at the beginning and the end, where the vehicle is not very far from the center of the intersection. Why?

11. The authors mentioned that experiment V is to investigate the effect of FCL. But in experiment IV, AF-UKF has already been studied. 

Author Response

Comments from Reviewer 2 and Responses

Comment: In this work, the authors propose a vehicle localization scheme based on sensor fusion of UWB/GPS/INS. To mitigate the effect of NLOS for UWB systems, the authors develop methods using ARIMA and fuzzy calibration logic. Most parts of the paper are well written. The authors need to address the following concerns in their revisions.

Response: Thank you very much for reviewing our manuscript very carefully. We have addressed the concerns in our revision as follows.

Comment: 1.Please explain what distinguishes this paper with the previous work [17]-[19].

Response: Many thanks for your insightful suggestion. Compared with the published work[17]~[19], this paper was concentrated on how to address NLOS problem and achieve lane-level positioning at urban intersections using low-cost sensors (MEMS-INS and commercial grade GPS). The previous work had not taken NLOS detection and mitigation of UWB into consideration. However, in vehicle localization scenarios, NLOS propagation of UWB may significantly deteriorate the positioning performance. We have proposed an ARIMA model to address this problem.  Besides, we develop the AF-UKF method to realize the global fusion of UWB with other low-cost onboard sensors. Continuous lane-level positioning is achieved in GPS-challenging environment through our strategy.

To clarify the difference between our paper with the previous work, we have modified the fifth paragraph of the introduction to make these distinctions clear to readers.

Comment: 2. The authors mentioned that the problem of NLOS identification and error mitigation has been studied for UWB in dense indoor environment, but not for outdoors. In general, indoor scenario is more challenging. Then why are the methods developed for indoors not applicable for outdoors?

Response: Thanks for your thorough consideration. It is our negligence for not explicating the real difficulties of using UWB to locate vehicles.

In general, indoor scenario is more challenging in terms of the high occurrence of NLOS and multipath propagation. However, indoor environment is relatively stable and indoor targets are usually motionless or slow-speed (compared with outdoors vehicle), those method developed for indoor environment may be not applicable for traffic scenarios. We encounter different challenges that associate with complex surroundings and high speed. For example, the signal propagation between the target vehicle and the roadside UWB is LOS, if there is a big vehicle passing by quickly and the propagation is obstructed, it may cause sudden transition from LOS to NLOS and then from NLOS to LOS. In dynamic traffic scenarios, we have to find a more appropriate method (namely ARIMA in this paper) to capture those sudden jumps of UWB and make the NLOS mitigation algorithm more suitable for vehicle applications.

   In the revised manuscript, we have changed some description in the 5th and the 9th paragraphs of introduction to make it clearer.

Comment: 3. Fix "trash environment" in page 3.

Response: Thanks for you carefully work and we have corrected the word “trash” into “harsh”.

Comment: 4. Please explain why the stationary assumption is essential for NLOS mitigation.

Response: It is perhaps our ambiguous description in the original paper that cause you confused. Whether the time series of UWB measurement is stationary or not does not affect the NLOS mitigation. The time series of UWB measurement is stationary when the vehicle is motionless. The ARMA model is suitable under this circumstance. When the vehicle starts to move, the time series of UWB measurement become nonstationary. In order to model nonstationary time series, we used the ARIMA method in this paper.

We have changed some description in the second paragraph of section 3 to avoid misunderstanding.

Comment: 5. Write the explicit mathematical form for the residual error Re(o;Sk) in (5)

Response: We have already denoted the explicit mathematical form for the residual error in the revised manuscript.

Comment: 6. What does "UWB bias error" mean in page 7.

Response: Sorry to cause you confused. UWB bias error is a constant bias term of UWB range error. We have modified the description in our manuscript.

Comment: 7. Denote the first 9 terms in [8] one by one.

Response: Sorry to cause you confused again. The first 9 terms in [8] are latitude, longitude, height, the velocity component along east direction, the velocity component along north direction, the velocity component along up direction, pitch angle, roll angle, azimuth angle respectively. We have already denoted the first 9 terms in [8] in revised manuscript.

Comment: 8. Is there any reference for the system transition model in (11)?

Response: We refer to a RISS model from the book named “Fundamentals of inertial navigation, satellite-based positioning and their integration”, we have already added it as the reference [30].

Comment: 9. Explain in practice how to get the covariance matrix for the measurement noise of GPS and UWB in (16)-(18).

Response: Thanks for your experienced advice. The preliminary measurement noise of GPS and UWB is acquired through priori knowledge and statistic trials. Before the experiment, we had carried out several static tests at the intersection to get range variance of UWB under LOS situation. Combining statistic results with priori knowledge, we get the preliminary covariance matrix of the measurement noise of GPS and UWB. In the actual implementation of UKF, some fine tuning of the preliminary covariance is conducted to achieve a better positioning performance.

We have added explanation of how to get the covariance matrix of the measurement noise in our paper.

Comment: 10. In Figure 10, the positioning error is very large at the beginning and the end, where the vehicle is not very far from the center of the intersection. Why?

Response: Thanks a lot for reviewing our manuscript very carefully. The positioning error at the beginning and the end is mainly caused by anomaly measurement from the NLOS nodes of UWB. In test III, when the vehicle is at the beginning and the end of the route, the roadside UWB nodes along the north-south direction is blocked by the trees or roadside infrastructures. The output of some seriously blocked NLOS nodes in test III is significantly biased and to some extent, can be regarded as abnormal measurement. Those abnormal measurement caused a large positioning error at the beginning and the end of the route.

Comment: 11. The authors mentioned that experiment V is to investigate the effect of FCL. But in experiment IV, AF-UKF has already been studied.

Response: Please accept our apologizes for making your confused. Both the experiment IV and V are designed to validate the performance of our fusion strategy (AF-UKF). We just want to employ different routes to make the experiment more convincing. In view of this, we keep the test IV in our paper. It seems not necessary.

 In fact, the trajectory V is a more comprehensive route, which includes straight line, left turn, right turn and “u” turn, therefore the conventional UKF method is investigated and compared with the AF-UKF in test V to evaluate the performance of FCL.

Reviewer 3 Report

The manuscript is well written, the design of the experiment is adequate and the results are convincing.

Author Response

Comments from Reviewer 3 and Responses

Comment: The manuscript is well written, the design of the experiment is adequate and the results are convincing.

Response: Thanks a lot for your positive and encouraging comments.

Reviewer 4 Report

The paper describes an UWB-based localization approach and it presents a model to address the issues related to NLOS.

The topic of the paper is interesting, but the presentation of the work should be really improved, both from the point of view of the use of english and of the layout. Moreover, it is not clear from the paper which are the novel contribution of authors. In particular, the use of the Kalman Filter seems not to be novel. 

The same font-size should be used throughout the paper. The same font should be used in the text and in formulas.

The caption of Fig.2 starts with “of”.

Please explain the meaning of attitude state.

English needs some revision, e.g.:

- “work“ —> workS (page 2, first line)

- “immunity of” —> immunity TO (page 2)

- “make it robust” —> makeS

- The sentence starting at line 183 should star with a capital letter.

- “In order to modeling“ —> to model

Presentation:

- Please insert a space before acronymes and before references.

- Never put a space before commas. Always use a space after commas.

Author Response

Comments from Reviewer 4 and Responses

Comment: The paper describes an UWB-based localization approach and it presents a model to address the issues related to NLOS. The topic of the paper is interesting, but the presentation of the work should be really improved, both from the point of view of the use of english and of the layout.

Response: Thanks a lot for reviewing our manuscript very carefully. We deeply appreciate your constructive comments that greatly help improve the presentation of this manuscript. We revised the manuscript in according with your comments, and carefully proof-read the manuscript to minimize typographical and grammatical errors.

Comment: Moreover, it is not clear from the paper which are the novel contribution of authors. In particular, the use of the Kalman Filter seems not to be novel.

Response: Please accept our apologies for not being able to present clearly the real contribution of the work. In this paper, we concentrate on achieving continuous lane-level positioning in GPS-challenging environment as urban intersections using UWB and low-cost onboard sensors. The major contributions of our work are the proposed ARIMA algorithms for UWB raw measurements preprocessing and the AF-UKF for global fusion:

1) In the practical employment of UWB, NLOS propagation may seriously degrade the positioning performance. Therefore, we introduce the ARIMA model to address NLOS problem. To our knowledge, previous studies have rarely investigated the NLOS detection and mitigation for land vehicle in practical traffic scenario.

2) To further mitigate NLOS and realize global fusion of UWB with other sensors, we develop the AF-UKF algorithm, in which the fuzzy calibration logic (See in the section 4.4) is designed. Compared with traditional UKF method, the AF-UKF can adaptively adjust the dependence on each received UWB measurement to further mitigate NLOS and multipath interferences. Ground test results show that the positioning accuracy is improved remarkably, e.g. over 37% (as shown in test V), through the AF-UKF. With the help of the elaborate fuzzy algorithm, the AF-UKF become an efficient and effective method for land vehicle positioning.

In order to make readers easier to get the novel contribution of our work, we have modified some description in abstract, introduction and conclusion to highlight the innovation points of our paper. Thanks again for you thorough consideration.

Comment: The same font-size should be used throughout the paper. The same font should be used in the text and in formulas.

Response: Thanks a lot for your advice. Many grammatical or typographical errors have been revised.

Comment: The caption of Fig.2 starts with “of”.

Response: Correction have been made in revised manuscript.

Comment: Please explain the meaning of attitude state.

Response: Sorry to make your confused about the attitude state in the state model. The position states of vehicle are latitude, longitude and height. The velocity states are the velocity component along east direction, the velocity component along north direction and the velocity component along up direction respectively. The attitude states denote pitch angle, roll angle and azimuth angle respectively. We have already denoted the first 9 terms in equation (8) in our paper.

Comment: English needs some revision, e.g.:

- “work“ —> workS (page 2, first line)

- “immunity of” —> immunity TO (page 2)

- “make it robust” —> makeS

- The sentence starting at line 183 should star with a capital letter.:

- “In order to modeling“ —> to model

Response: Thanks for your carefully work. It helps us improve the English writing of the manuscript greatly. We also have our manuscript checked by an experienced English speaking colleague to improve the English editing. The suggested correction had been made in revised manuscript.

Round  2

Reviewer 2 Report

The authors addressed all my comments. 

Reviewer 4 Report

I appreciate the efforts made by authors to improve the quality of the manuscript.

However, additional revision of English needs to be performed.